# RRLS : ROBUST REINFORCEMENT LEARNING SUITE

## ABSTRACT

Robust reinforcement learning is the problem of learning control policies that provide optimal worst-case performance against a span of adversarial environments. It is a crucial ingredient for deploying algorithms in real-world scenarios with prevalent environmental uncertainties and has been a long-standing object of attention in the community, without a standardized set of benchmarks. This contribution endeavors to fill this gap. We introduce the Robust Reinforcement Learning Suite (RRLS), a benchmark suite based on Mujoco environments. RRLS provides six continuous control tasks with two types of uncertainty sets for training and evaluation. Our benchmark aims to standardize robust reinforcement learning tasks, facilitating reproducible and comparable experiments, in particular those from recent state-of-the-art contributions, for which we demonstrate the use of RRLS. It is also designed to be easily expandable to new environments.

## 1 INTRODUCTION

Reinforcement learning (RL) algorithms frequently encounter difficulties in maintaining performance when confronted with dynamic uncertainties and varying environmental conditions. This lack of robustness significantly limits their applicability in the real world. Robust reinforcement learning addresses this issue by focusing on learning policies that ensure optimal worst-case performance across a range of adversarial conditions. For instance, an aircraft control policy should be capable of effectively managing various configurations and atmospheric conditions without requiring retraining. This is critical for applications where safety and reliability are paramount to avoid a drastic decrease in performance Morimoto & Doya (2005); Tessler et al. (2019).

The concept of robustness, as opposed to resilience, places greater emphasis on maintaining performance without further training. In robust reinforcement learning (RL), the objective is to optimize policies for the worst-case scenarios, ensuring that the learned policies can handle the most challenging conditions. This framework is formalized through robust Markov decision processes (MDPs), where the transition dynamics are subject to uncertainties. Despite significant advancements in robust RL algorithms, the field lacks standardized benchmarks for evaluating these methods. This hampers reproducibility and comparability of experimental results (Moos et al., 2022). To address this gap, we introduce the Robust Reinforcement Learning Suite, a comprehensive benchmark suite designed to facilitate rigorous evaluation of robust RL algorithms.

The Robust Reinforcement Learning Suite (RRLS) provides six continuous control tasks based on Mujoco Todorov et al. (2012) environments, each with distinct uncertainty sets for training and evaluation. By standardizing these tasks, RRLS enables reproducible and comparable experiments, promoting progress in robust RL research. The suite includes four compatible baselines with the RRLS benchmark, which are evaluated in static environments to demonstrate their efficacy. In summary, our contributions are the following :

- Our first contribution aims to establish a standardized benchmark for robust RL, addressing the critical need for reproducibility and comparability in the field (Moos et al., 2022). The RRLS benchmark suite represents a significant step towards achieving this goal, providing a robust framework for evaluating state-of-the-art robust RL algorithms.

- Our second contribution is a comparison and evaluation of different Deep Robust RL algorithms in Section 5 on our benchmark, showing the pros and cons of different methods.

## 2 PROBLEM STATEMENT

**Reinforcement learning.** Reinforcement Learning (RL) (Sutton & Barto, 2018) addresses the challenge of developing a decision-making policy for an agent interacting with a dynamic environment over multiple time steps. This problem is modeled as a Markov Decision Process (MDP) (Puterman, 2014) represented by the tuple $(S, A, p, r)$, which includes states $S$, actions $A$, a transition kernel $p(s_{t+1}|s_t, a_t)$, and a reward function $r(s_t, a_t)$. For simplicity, we assume a unique initial state $s_0$, though the results generalize to an initial state distribution $p_0(s)$. A stationary policy $\pi(s) \in \Delta_A$ maps states to distributions over actions. The objective is to find a policy $\pi$ that maximizes the expected discounted return

$$J^\pi = \mathbb{E}_{s_0 \sim \rho}[v_p^\pi(s_0)] = \mathbb{E}\Big[\sum_{t=0}^\infty \gamma^t r(s_t, a_t)|a_t \sim \pi, s_{t+1} \sim p, s_0 \sim \rho\Big], \tag{1}$$

where $v_p^\pi$ is the value function of $\pi$, $\gamma \in [0, 1)$ is the discount factor, and $s_0$ is drawn from the initial distribution $\rho$. The value function $v_p^\pi$ of policy $\pi$ assigns to each state $s$ the expected discounted sum of rewards when following $\pi$ starting from $s$ and following transition kernel $p$. An optimal policy $\pi^*$ maximizes the value function in all states. To converge to the (optimal) value function, the value iteration (VI) algorithm can be applied, which consists in repeated application of the (optimal) Bellman operator $T^*$ to value functions:

$$v_{n+1}(s) = T^* v_n(s) := \max_{\pi(s) \in \Delta_A} \mathbb{E}_{a \sim \pi(s)}[r(s, a) + \mathbb{E}_p[v_n(s')]]. \tag{2}$$

Finally, the $Q$ function is also defined similarly to Equation equation 1 but starting from specific state/action $(s, a)$ as $\forall(s, a) \in S \times A$:

$$Q^\pi(s, a) = \mathbb{E}\Big[\sum_{t=0}^\infty \gamma^t r(s_t, a_t)|a_t \sim \pi, s_{t+1} \sim p, s_0 = s, a_0 = a\Big]. \tag{3}$$

**Robust reinforcement learning.** In a Robust MDP (RMDP) Iyengar (2005); Nilim & El Ghaoui (2005), the transition kernel $p$ is not fixed and can be chosen adversarially from an uncertainty set $\mathcal{P}$ at each time step. The pessimistic value function of a policy $\pi$ is defined as $v_{\mathcal{P}}^\pi(s) = \min_{p \in \mathcal{P}} v_p^\pi(s)$. An optimal robust policy maximizes the pessimistic value function $v_{\mathcal{P}}$ in any state, leading to a $\max_\pi \min_p$ optimization problem. This is known as the static model of transition kernel uncertainty, as $\pi$ is evaluated against a static transition model $\pi$. Robust Value Iteration (RVI) (Iyengar, 2005; Wiesemann et al., 2013) addresses this problem by iteratively computing the one-step lookahead best pessimistic value:

$$v_{n+1}(s) = T_{\mathcal{P}}^* v_n(s) := \max_{\pi(s) \in \Delta_A} \min_{p \in \mathcal{P}} \mathbb{E}_{a \sim \pi(s)}[r(s, a) + \mathbb{E}_p[v_n(s')]]. \tag{4}$$

This dynamic programming formulation is called the dynamic model of transition kernel uncertainty, as the adversary picks the next state distribution only for the current state-action pair, after observing the current state and the agent's action at each time step (and not a full transition kernel). The $T_{\mathcal{P}}^*$ operator, known as the robust Bellman operator, ensures that the sequence of $v_n$ functions converges to the robust value function $v_{\mathcal{P}}^*$, provided the adversarial transition kernel belongs to the simplex of $\Delta_S$ and that the static and dynamic cases have the same solutions for stationary agent policies Iyengar (2022).

**Robust reinforcement learning as a two-player game.** Robust MDPs can be represented as zero-sum two-player Markov games (Littman, 1994; Tessler et al., 2019) where $\bar{S}, \bar{A}$ are respectively the state and action set of the adversarial player. In a zero-sum Markov game, the adversary tries to minimize the reward or maximize $-r$. Writing $\bar{\pi} : \bar{S} \to \bar{A} := \Delta_S$ the policy of this adversary, the robust MDP problem turns to $\max_\pi \min_{\bar{\pi}} v^{\pi, \bar{\pi}}$, where $v^{\pi, \bar{\pi}}(s)$ is the expected sum of discounted rewards obtained when playing $\pi$ (agent actions) against $\bar{\pi}$ (transition models) at each time step from $s$. In the specific case of robust RL as a two player-game, $\bar{S} = S \times A$. This enables introducing the robust value iteration sequence of functions

$$v_{n+1}(s) := T^{**} v_n(s) := \max_{\pi(s) \in \Delta_A} \min_{\bar{\pi}(s, a) \in \Delta_S} (T^{\pi, \bar{\pi}} v_n)(s) \tag{5}$$

where $T^{\pi, \bar{\pi}} := \mathbb{E}_{a \sim \pi(s)}[r(s, a) + \gamma \mathbb{E}_{s' \sim \bar{\pi}(s, a)} v_n(s')]$ is a zero-sum Markov game operator. These operators are also $\gamma-$contractions to their respective fixed point $v^{\pi, \bar{\pi}}$ and $v^{**} = v_{\mathcal{P}}^*$ Tessler et al. (2019). This two-player game formulation will be used in the evaluation of the RRLS in Section 5.

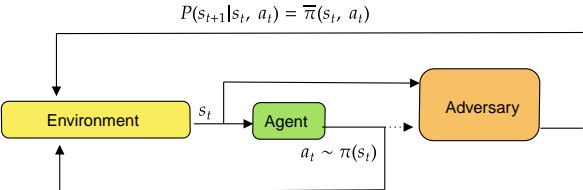

$$P(s_{t+1}|s_t, a_t) = \overline{\pi}(s_t, a_t)$$

Figure 1: Relation between Robust RL and Zero-sum Markov Game

## 3 RELATED WORKS

### 3.1 REINFORCEMENT LEARNING BENCHMARK

The landscape of reinforcement learning (RL) benchmarks has evolved significantly, enabling the accelerated development of RL algorithms. Prominent among these benchmarks are the Atari Arcade Learning Environment (ALE) Bellemare et al. (2012), OpenAI Gym Brockman et al. (2016), more recently Gymnasium Towers et al. (2023), and the DeepMind Control Suite (DMC) Tassa et al. (2018). The aforementioned benchmarks have established standardized environments for the evaluation of RL agents across discrete and continuous action spaces, thereby fostering the reproducibility and comparability of experimental results. The ALE has been particularly influential, offering a diverse set of Atari games that have become a standard testbed for discrete control tasks Bellemare et al. (2012). Moreover, the OpenAI Gym extended this approach by providing a more flexible and extensive suite of environments for various RL tasks, including discrete and continuous control Brockman et al. (2016). Similarly, the DMC Suite has been essential for benchmarking continuous control algorithms, offering a set of challenging tasks that facilitate evaluating algorithm performance Tassa et al. (2018). In addition to these general-purpose benchmarks, specialized benchmarks have been developed to address specific research needs. For instance, the DeepMind Lab focuses on 3D navigation tasks from pixel inputs Beattie et al. (2016), while ProcGen Cobbe et al. (2019) offers procedurally generated environments to evaluate the generalization capabilities of RL agents. The D4RL benchmark targets offline RL methods by providing datasets and tasks specifically designed for offline learning scenarios Fu et al. (2021), and RL Unplugged Gulcehre et al. (2020) offers a comprehensive suite of benchmarks for evaluating offline RL algorithms. RL benchmarks such as Meta-World Yu et al. (2021) have been developed to evaluate the ability of RL agents to transfer knowledge across multiple tasks. Meta-World provides a suite of robotic manipulation tasks designed to test RL algorithms' adaptability and generalization in multitask learning scenarios. Similarly, RLBench James et al. (2020) offers a variety of tasks for robotic learning, focusing on the performance of RL agents in multi-task settings. Recent contributions such as the Unsupervised Reinforcement Learning Benchmark (URLB) Lee et al. (2021) have further expanded the scope of RL benchmarks by targeting unsupervised learning methods. URLB aims to accelerate progress in unsupervised RL by providing a suite of environments and baseline implementations, promoting algorithm development that does not rely on labeled data for training. Additionally, the CoinRun benchmark Cobbe et al. (2020) and Sonic Benchmark Nichol et al. (2018) focus on evaluating generalization and transfer learning in RL through procedurally generated levels and video game environments, respectively. Finally, benchmarks like the Behavior Suite (bsuite) Osband et al. (2019) have been designed to test specific capabilities of RL agents, such as memory, exploration, and generalization. Closer to our work, safety in RL is another critical area where benchmarks like SafetyGym Achiam & Amodei (2019) have been instrumental. SafetyGym evaluates how well RL agents can perform tasks while adhering to safety constraints, which is crucial for real-world applications where safety cannot be compromised. Despite the progress in benchmarking RL algorithms, there has been a notable gap in benchmarks specifically designed for robust RL, which aims to learn policies that perform optimally in the worst-case scenario against adversarial environments. This gap highlights the need for standardized benchmarks (Moos et al., 2022) that facilitate reproducible and comparable experiments in robust RL. In the next section, we introduce existing robust RL algorithms.

## 3.2 ROBUST REINFORCEMENT LEARNING ALGORITHMS

Two principal classes of practical, robust reinforcement learning algorithms exist, those that can interact solely with a nominal transition kernel (or center of the uncertainty set), and those that can sample from the entire uncertainty ball. While the former is more mathematically founded, it is unable to exploit transitions that are not sampled from the nominal kernel and consequently exhibits lower performance. In this benchmark, only the Deep Robust RL as two-player games that use samples from the entire uncertainty set are implemented.

**Nominal-based Robust/risk-averse algorithms.** The idea of this class of algorithms is to approximate the inner minimum operator present robust Bellman operator in Equation equation 4. Previous work has typically employed a dual approach to the minimum problem, whereby the transition probability is constrained to remain within a specified ball around the nominal transition kernel. Practically, robustness is equivalent to regularization (Derman et al., 2021) and for example the SAC algorithm Haarnoja et al. (2018) has been shown to be robust due to entropic regularization. In this line of work, (Kumar et al., 2022) derived approximate algorithm for RMPDS with $L_p$ balls, (Clavier et al., 2022) for $\chi^2$ constrain and (Liu et al., 2022) for KL divergence. Finally, Wang et al. (2023) proposes a novel online approach to solve RMDP. Unlike previous works that regularize the policy or value updates, Wang et al. (2023) achieves robustness by simulating the worst kernel scenarios for the agent while using any classical RL algorithm in the learning process. These Robust RL approaches have received recent theoretical attention, from a statistical point of view (sample complexity) (Yang et al., 2022; Panaganti & Kalathil, 2022; Clavier et al., 2023; Shi et al., 2024) as well as from an optimization point of view (Grand-Clément & Kroer, 2021), but generally do not directly translate to algorithms that scale up to complex evaluation benchmarks.

**Deep Robust RL as two-player games.** A common approach to solving robust RL problems is cast the optimization process as a two-player game, as formalized by Morimoto & Doya (2005), described in Section 2, and summarized in Figure 1. In this framework, an adversary, denoted by $\bar{\pi} : \mathcal{S} \times \mathcal{A} \to \mathcal{P}$, is introduced, and the game is formulated as

$$\max_{\pi} \min_{\bar{\pi}} \mathbb{E}\left[\sum_{t=0}^{\infty} \gamma^t r(s_t, a_t, s_{t+1}) | s_0, a_t \sim \pi(s_t), p_t = \bar{\pi}(s_t, a_t), s_{t+1} \sim p_t(\cdot | s_t, a_t)\right].$$

Most methods differ in how they constrain $\bar{\pi}$'s action space within the uncertainty set. A first family of methods define $\bar{\pi}(s_t) = p_{ref} + \Delta(s_t)$, where $p_{ref}$ denotes the reference (nominal) transition function. Among this family, Robust Adversarial Reinforcement Learning (RARL) (Pinto et al., 2017) applies external forces at each time step $t$ to disturb the reference dynamics. For instance, the agent controls a planar monopod robot, while the adversary applies a 2D force on the foot. In noisy action robust MDPs (NR-MDP) (Tessler et al., 2019) the adversary shares the same action space as the agent and disturbs the agent's action $\pi(s)$. Such gradient-based approaches incur the risk of finding stationary points for $\pi$ and $\bar{\pi}$ which do not correspond to saddle points of the robust MDP problem. To prevent this, Mixed-NE (Kamalaruban et al., 2020) defines mixed strategies and uses stochastic gradient Langevin dynamics. Similarly, Robustness via Adversary Populations (RAP) (Vinitsky et al., 2020) introduces a population of adversaries, compelling the agent to exhibit robustness against a diverse range of potential perturbations rather than a single one, which also helps prevent finding stationary points that are not saddle points.

Aside from this first family, State Adversarial MDPs (Zhang et al., 2020; 2021; Stanton et al., 2021) involve adversarial attacks on state observations, which implicitly define a partially observable MDP. This case aims not to address robustness to the worst-case transition function but rather against noisy, adversarial observations.

A third family of methods considers the general case of $\bar{\pi}(s_t, a_t) = p_t$ or $\bar{\pi}(s_t) = p_t$, where $p_t \in \mathcal{P}$. Minimax Multi-Agent Deep Deterministic Policy Gradient (M3DDPG) (Li et al., 2019) is designed to enhance robustness in multi-agent reinforcement learning settings but boils down to standard robust RL in the two-agents case. Max-min TD3 (M2TD3) (Tanabe et al., 2022) considers a policy $\pi$, defines a value function $Q(s, a, p)$ which approximates $Q_p^{\pi}(s, a) = \mathbb{E}_{s' \sim p}[r(s, a, s') + \gamma V_p^{\pi}(s')]$, updates an adversary $\bar{\pi}$ so as to minimize $Q(s, \pi(s), \bar{\pi}(s))$ by taking a gradient step with respect to $\bar{\pi}$'s parameters, and updates the policy $\pi$ using a TD3 gradient update in the direction maximizing $Q(s, \pi(s), \bar{\pi}(s))$. As such, M2TD3 remains a robust value iteration method that solves the dynamic

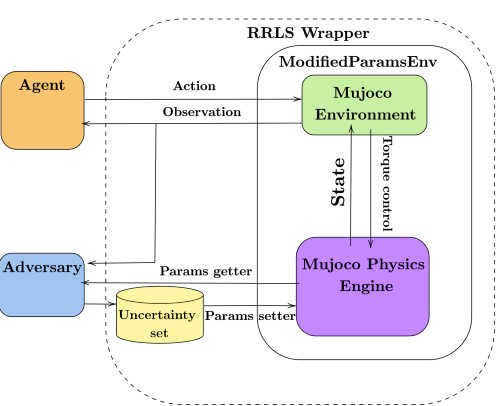

Figure 2: RRLS architecture

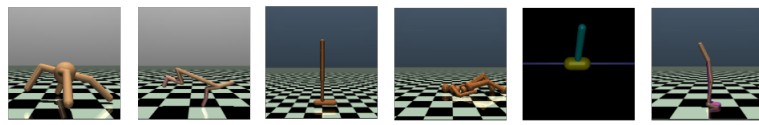

Figure 3: Visual representation of various reinforcement learning environments including **Ant**, **HalfCheetah**, **Hopper**, **Humanoid Stand Up**, **Inverted Pendulum**, and **Walker**.

problem by alternating updates on $\pi$ and $\bar{\pi}$, but since it approximates $Q_p^\pi$, it is also closely related to the method we introduce in the next section.

**Domain randomization.** Domain randomization (DR) (Tobin et al., 2017) learns a value function $V(s) = \max_\pi \mathbb{E}_{p\sim\mathcal{U}(\mathcal{P})} V_p^\pi(s)$ which maximizes the expected return *on average* across a fixed distribution on $\mathcal{P}$. As such, DR approaches do not optimize the worst-case performance. Nonetheless, DR has been used convincingly in applications (Mehta et al., 2020; OpenAI et al., 2019). Similar approaches also aim to refine a base DR policy for application to a sequence of real-world cases (Lin et al., 2020; Dennis et al., 2020; Yu et al., 2018). For a more complete survey of recent works in robust RL, we refer the reader to the work of Moos et al. (2022).

## 4 RRLS: BENCHMARK ENVIRONMENTS FOR ROBUST RL

This section introduces the Robust Reinforcement Learning Suite, which extends the Gymnasium Towers et al. (2023) API with two additional methods: `set_params` and `get_params`. These methods are integral to the `ModifiedParamsEnv` interface, facilitating environment parameter modifications within the benchmark environment. Typically, these methods are used within a wrapper to simplify parameter modifications during evaluation. In the RRLS architecture (Figure 2), the adversary begins by retrieving parameters from the uncertainty set and setting them in the environment using the `ModifiedParamsEnv` interface. The agent then acts based on the current state of the environment, and the Mujoco Physics Engine updates the state accordingly. The agent observes this updated state, completing the interaction loop. Multiple MuJoCo environments are provided (Figure 3), each with a two default uncertainty sets, inspired respectively by those used in the experiments of RARL (Pinto et al., 2017) (Table 1) and M2TD3 (Tanabe et al., 2022) (Table 2). This variety allows for a comprehensive evaluation of robust RL algorithms, ensuring that the benchmarks encompass a wide range of scenarios.

Several MuJoCo environments are proposed, each with distinct action and observation spaces. Figure 3 shows a visual representation of all provided environments. In all environments, the observation space corresponds to the positional values of various body parts followed by their velocities, with all positions listed before all velocities. The environments are as follows:

- **Ant**: A 3D robot with one torso and four legs, each with two segments. The goal is to move forward by coordinating the legs and applying torques on the eight hinges. The action dimension is 8, and the observation dimension is 27.

- **HalfCheetah**: A 2D robot with nine body parts and eight joints, including two paws. The goal is to run forward quickly by applying torque to the joints. Positive rewards are given for forward movement, and negative rewards for moving backward. The action dimension is 6, and the observation dimension is 17.

- **Hopper**: A 2D one-legged figure with four main parts: torso, thigh, leg, and foot. The goal is to hop forward by applying torques on the three hinges. The action dimension is 3, and the observation dimension is 11.

- **Humanoid Stand Up**: A 3D bipedal robot resembling a human, with a torso, legs, and arms, each with two segments. The environment starts with the humanoid lying on the ground. The goal is to stand up and remain standing by applying torques to the various hinges. The action dimension is 17, and the observation dimension is 376.

- **Inverted Pendulum**: A cart that can move linearly, with a pole fixed at one end. The goal is to balance the pole by applying forces to the cart. The action dimension is 1, and the observation dimension is 4.

- **Walker**: A 2D two-legged figure with seven main parts: torso, thighs, legs, and feet. The goal is to walk forward by applying torques on the six hinges. The action dimension is 6, and the observation dimension is 17.

The RRLS architecture enables parameter modifications and adversarial interactions using the `gymnasium` Towers et al. (2023) interface. The `set_params` and `get_params` methods in the `ModifiedParamsEnv` interface directly access and modify parameters in the Mujoco Physics Engine. All modifiable parameters are listed in Appendix A and lie in the uncertainty set described below.

**Uncertainty Sets.** Non-rectangular uncertainty sets (opposed to rectangular ones as defined in (Iyengar, 2005)) are proposed based on MuJoCo environments, detailed in Table 1. These sets, based on previous work evaluating M2TD3 Tanabe et al. (2022) and RARL Pinto et al. (2017), ensure thorough testing of robust RL algorithms under diverse conditions. For instance, the uncertainty range for the torso mass in the HumanoidStandUp 2 and 3 environments spans from 0.1 to 16.0 (Table 1), ensuring challenging evaluation of RL methods. Three uncertainty sets—1D, 2D, and 3D—are provided for each environment, ranging from simple to challenging.

Table 1: List of parameters uncertainty sets based on M2TD3 in RRLS

| Environment | Uncertainty set $\mathcal{P}$ | Reference values | Uncertainty parameters |
|---|---|---|---|
| Ant 1 | $[0.1, 3.0]$ | 0.33 | torsomass |
| Ant 2 | $[0.1, 3.0] \times [0.01, 3.0]$ | (0.33, 0.04) | torso mass; front left leg mass |
| Ant 3 | $[0.1, 3.0] \times [0.01, 3.0] \times [0.01, 3.0]$ | (0.33, 0.04, 0.06) | torso mass; front left leg mass; front right leg mass |
| HalfCheetah 1 | $[0.1, 3.0]$ | 0.4 | world friction |
| HalfCheetah 2 | $[0.1, 4.0] \times [0.1, 7.0]$ | (0.4, 6.36) | world friction; torso mass |
| HalfCheetah 3 | $[0.1, 4.0] \times [0.1, 7.0] \times [0.1, 3.0]$ | (0.4, 6.36, 1.53) | world friction; torso mass; back thigh mass |
| Hopper 1 | $[0.1, 3.0]$ | 1.00 | world friction |
| Hopper 2 | $[0.1, 3.0] \times [0.1, 3.0]$ | (1.00, 3.53) | world friction; torso mass |
| Hopper 3 | $[0.1, 3.0] \times [0.1, 3.0] \times [0.1, 4.0]$ | (1.00, 3.53, 3.93) | world friction; torso mass; thigh mass |
| HumanoidStandup 1 | $[0.1, 16.0]$ | 8.32 | torsomass |
| HumanoidStandup 2 | $[0.1, 16.0] \times [0.1, 8.0]$ | (8.32, 1.77) | torso mass; right foot mass |
| HumanoidStandup 3 | $[0.1, 16.0] \times [0.1, 5.0] \times [0.1, 8.0]$ | (8.32, 1.77, 4.53) | torso mass; right foot mass; left thigh mass |
| InvertedPendulum 1 | $[1.0, 31.0]$ | 4.90 | polemass |
| InvertedPendulum 2 | $[1.0, 31.0] \times [1.0, 11.0]$ | (4.90, 9.42) | pole mass; cart mass |
| Walker 1 | $[0.1, 4.0]$ | 0.7 | world friction |
| Walker 2 | $[0.1, 4.0] \times [0.1, 5.0]$ | (0.7, 3.53) | world friction; torso mass |
| Walker 3 | $[0.1, 4.0] \times [0.1, 5.0] \times [0.1, 6.0]$ | (0.7, 3.53, 3.93) | world friction; torso mass; thigh mass |

RRLS also directly provides the uncertainty sets from the RARL (Pinto et al., 2017) paper. These sets apply destabilizing forces at specific points in the system, encouraging the agent to learn robust control policies.

**Wrappers.** We introduce environment wrappers to facilitate the implementation of various deep robust RL baselines such as M2TD3 Tanabe et al. (2022), RARL Pinto et al. (2017), Domain Randomization Tobin et al. (2017), NR-MDP Tessler et al. (2019) and all algorithms deriving from

Table 2: List of parameters uncertainty sets based on RARL in RRLS

| Environment | Uncertainty set $\mathcal{P}$ | Uncertainty parameters |
|---|---|---|
| Ant Rarl | $[-3.0, 3.0]^{\times 6}$ | torso force x; torso force y; front left leg force x; front left leg force y; front right leg force x; front right leg force y |
| HalfCheetah Rarl | $[-3.0, 3.0]^{\times 6}$ | torso force x; torso force y; back foot force x; back foot force y; forward foot force x; forward foot force y |
| Hopper Rarl | $[-3.0, 3.0]^{\times 2}$ | foot force x; foot force y |
| HumanoidStandup Rarl | $[-3.0, 3.0]^{\times 6}$ | torso force x; torso force y; right thigh force x; right thigh force y; left foot force x; left foot force y |
| InvertedPendulum Rarl | $[-3.0, 3.0]^{\times 2}$ | pole force x; pole force y |
| Walker Rarl | $[-3.0, 3.0]^{\times 4}$ | leg force x; leg force y; left foot force x; left foot force y |

Robust Value Iteration, ensuring researchers can easily apply and compare different methods within a standardized framework. The wrappers are described as follows:

- The `ModifiedParamsEnv` interface includes methods `set_params` and `get_params`, which modifying and retrieving environment parameters. This interface allows dynamic adjustment of the environment during training or evaluation.

- The `DomainRandomization` wrapper enables domain randomization by sampling environment parameters from the uncertainty set between episodes. It wraps an environment following the `ModifiedParamsEnv` interface and uses a randomization function to draw new parameter sets. If no function is set, the parameter is sampled uniformly. Parameters reset at the beginning of each episode, ensuring diverse training conditions.

- The `Adversarial` wrapper converts an environment into a robust reinforcement learning problem modeled as a zero-sum Markov game. It takes an uncertainty set and the `ModifiedParamsEnv` as input. This wrapper extends the action space to include adversarial actions, allowing for modifications of transition kernel parameters within a specified uncertainty set. It is suitable for reproducing robust reinforcement learning approaches based on adversarial perturbation in the transition kernel, such as RARL.

- The `ProbabilisticActionRobust` wrapper defines the adversary's action space as the same action space as the agent. The final action applied in the environment is a convex sum between the agent's action and the adversary's action: $a_{pr} = \alpha a + (1 - \alpha)\bar{a}$. The adversarial action's effect is bounded by the environment's action space, allowing the implementation of robust reinforcement learning methods around a reference transition kernel, such as NR-MDP or RAP.

**Evaluation Procedure.** Evaluating Robust Reinforcement Learning algorithms can feature a large variability in outcome statistics depending on a number of minor factors (such as random seeds, initial state, or collection of evaluation transition models). To address this, we propose a systematic approach using a function called `generate_evaluation_set`. This function takes an uncertainty set as input and returns a list of evaluation environments. In the static case, where the transition kernel remains constant across time steps, the evaluation set consists of environments spanned by a uniform mesh over the parameters set. The agent runs multiple trajectories in each environment to ensure comprehensive testing. Each dimension of the uncertainty set is divided by a parameter named `nb_mesh_dim`. This parameter controls the granularity of the evaluation environments. To standardize the process, we provide a default evaluation set for each uncertainty set (Table 1). This set allows for worst-case performance and average-case performance evaluation in static conditions.

## 5 BENCHMARKING ROBUST RL ALGORITHMS

**Experimental setup.** This section evaluates several baselines in static and dynamic settings using RRLS. We conducted experimental validation by training policies in the Ant, HalfCheetah, Hopper, HumanoidStandup, and Walker environments. We selected five baseline algorithms: TD3, Domain Randomization (DR), NR-MDP, RARL, and M2TD3. We select the most challenging scenarios, the 3D uncertainty set defined in Table 1, normalized between $[0, 1]^3$. For static evaluation, we used the standard evaluation procedure proposed in the previous section. Performance metrics were gathered after five million steps to ensure a fair comparison after convergence. All baselines were constructed using TD3 with a consistent architecture across all variants. The results were obtained by averaging over ten distinct random seeds. Appendices B, D.1, D.2, and D.3 provide further details on hyperparameters, network architectures, implementation choices, and training curves.

**Static worst-case performance.** Tables 3 and 4 report normalized scores for each method, averaged across 10 random seeds and 5 episodes per seed, for each transition kernel in the evaluation uncertainty set. To compare metrics across environments, the score $v$ of each method was normalized relative to the reference score of TD3. TD3 was trained on the environment using the reference transition kernel, and its score is denoted as $v_{TD3}$. The M2TD3 score, $v_{M2TD3}$, was used as the comparison target. The formula used to get a normalized score is $(v - v_{TD3})/(|v_{M2TD3} - v_{TD3}|)$. This defines $v_{TD3}$ as the minimum baseline and $v_{M2TD3}$ as the target. This standardization provides a metric that quantifies the improvement of each method over TD3 relative to the improvement of M2TD3 over TD3. Non-normalized results are available in Appendix C. As expected, M2TD3, RARL and DR perform better in terms of worst-case performance, than vanilla TD3. Surprisingly, RARL is outperformed by DR except for HalfCheetah, Hopper, and Walker in worst-case performance. Finally, M2TD3, which is a state-of-the-art algorithm, outperforms all baselines except on HalfCheetah where DR achieves a slightly, non-statistically significant, better score. One potential explanation for the superior performance of DR over robust reinforcement learning methods in the HalfCheetah environment is that the training of a conservative value function is not necessary. The HalfCheetah environment is inherently well-balanced, even with variations in mass or friction. Consequently, robust training, which typically aims to handle worst-case scenarios, becomes less critical. This insight aligns with the findings of Moskovitz et al. (2021), who observed similar results in this specific environment. The variance in the evaluations also needs to be addressed. In many environments, high variance prevents drawing statistical conclusions. For instance, HumanoidStandup shows a variance of 3.32 for M2TD3, complicating reliable performance assessments. Similar issues arise with DR in the same environment, showing a variance of 4.1. Such variances highlight the difficulty of making definitive comparisons across different robust reinforcement learning methods in these settings.

Table 3: Avg. of normalized static worst-case performance over 10 seeds for each method

|  | Ant | HalfCheetah | Hopper | HumanoidStandup | Walker | Average |
|---|---|---|---|---|---|---|
| TD3 | $0.0 \pm 0.34$ | $0.0 \pm 0.06$ | $0.0 \pm 0.21$ | $0.0 \pm 2.27$ | $0.0 \pm 0.1$ | $0.0 \pm 0.6$ |
| DR | $0.06 \pm 0.16$ | $\mathbf{1.07 \pm 0.36}$ | $0.86 \pm 0.82$ | $0.04 \pm 4.1$ | $0.57 \pm 0.37$ | $0.52 \pm 1.16$ |
| M2TD3 | $\mathbf{1.0 \pm 0.27}$ | $1.0 \pm 0.16$ | $\mathbf{1.0 \pm 0.65}$ | $\mathbf{1.0 \pm 3.32}$ | $\mathbf{1.0 \pm 0.63}$ | $\mathbf{1.0 \pm 1.01}$ |
| RARL | $0.44 \pm 0.3$ | $0.13 \pm 0.08$ | $0.5 \pm 0.22$ | $0.44 \pm 2.94$ | $0.12 \pm 0.09$ | $0.33 \pm 0.73$ |
| NR-MDP | $-0.25 \pm 0.1$ | $-0.10 \pm 0.24$ | $-0.31 \pm 0.4$ | $-2.22 \pm 1.51$ | $-0.04 \pm 0.01$ | $-0.58 \pm 0.45$ |

**Static average performance.** Similarly to the worst-case performance described above, average scores across a uniform distribution on the uncertainty set are reported in Table 4. While robust policies explicitly optimize for the worst-case circumstances, one still desires that they perform well across all environments. A sound manner to evaluate this is to average their scores across a distribution of environments. First, one can observe that DR outperforms the other algorithms. This was expected since DR is specifically designed to optimize the policy on average across a (uniform) distribution of environments. One can also observe that RARL performs worse on average than a standard TD3 in most environments (except HumanoidStandup), despite having better worst-case scores. This exemplifies how robust RL algorithms can output policies that lack applicability in practice. Finally, M2TD3 is still better than TD3 on average, and hence this study confirms that it optimizes for worst-case performance while preserving the average score.

Table 4: Avg. of normalized static average case performance over 10 seeds for each method

|  | Ant | HalfCheetah | Hopper | HumanoidStandup | Walker | Average |
|---|---|---|---|---|---|---|
| TD3 | $0.0 \pm 0.49$ | $0.0 \pm 0.22$ | $0.0 \pm 0.83$ | $0.0 \pm 1.36$ | $0.0 \pm 0.51$ | $0.0 \pm 0.68$ |
| DR | $\mathbf{1.65 \pm 0.05}$ | $\mathbf{2.31 \pm 0.27}$ | $\mathbf{2.08 \pm 0.49}$ | $\mathbf{1.15 \pm 2.47}$ | $\mathbf{1.22 \pm 0.34}$ | $\mathbf{1.68 \pm 0.72}$ |
| M2TD3 | $1.0 \pm 0.11$ | $1.0 \pm 0.19$ | $1.0 \pm 0.55$ | $1.0 \pm 1.43$ | $1.0 \pm 0.65$ | $1.0 \pm 0.59$ |
| RARL | $0.69 \pm 0.13$ | $-1.3 \pm 0.54$ | $-0.99 \pm 0.11$ | $0.47 \pm 1.92$ | $-0.35 \pm 0.83$ | $-0.3 \pm 0.71$ |
| NR-MDP | $0.44 \pm 0.03$ | $-0.58 \pm 0.17$ | $-0.85 \pm 0.001$ | $-0.83 \pm 0.24$ | $-1.08 \pm 0.01$ | $-0.58 \pm 0.15$ |

**Dynamic adversaries.** While the static and dynamic cases of transition kernel uncertainty lead to the same robust value functions in the idealized framework of rectangular uncertainty sets, most real-life situations (such as those in RRLS) fall short of this rectangularity assumption. Consequently, Robust Value Iteration algorithms, which train an adversarial policy $\bar{\pi}$ (whether they store it or not) might possibly lead to a policy that differs from those which optimize for the original $\max_\pi \min_p$ problem

introduced in Section 2. RRLS permits evaluating this feature by running rollouts of agent policies versus their adversaries, after optimization. RARL and NR-MDP simultaneously train a policy $\pi$ and an adversary $\bar{\pi}$. The policy is evaluated against its adversary over ten episodes. Observations in Table 5 demonstrate how RRLS can be used to compare RARL and NR-MDP against their respective adversaries, in raw score. However, this comparison should not be interpreted as a dominance of one algorithm over the other, since the uncertainty sets they are trained upon are not the same.

Table 5: Comparison of RARL and NR-MDP across different environments

| Method | HumanoidStandup ($10^4$) | Ant ($10^3$) | HalfCheetah ($10^2$) | Hopper ($10^3$) | Walker ($10^3$) |
|---|---|---|---|---|---|
| RARL | $9.84 \pm 3.36$ | $2.90 \pm 0.70$ | $-0.74 \pm 6.69$ | $1.04 \pm 0.16$ | $3.45 \pm 1.13$ |
| NR-MDP | $9.37 \pm 0.14$ | $5.58 \pm 0.64$ | $109.90 \pm 4.74$ | $3.14 \pm 0.53$ | $5.17 \pm 0.89$ |

**Training curves.** Figure 4 reports training curves for TD3, DR, RARL, and M2TD3 on the Walker environment, using RRLS (results for all other environments in Appendix B). Each agent was trained for 5 million steps, with cumulative rewards monitored over trajectories of 1,000 steps. Scores were averaged over 10 different seeds. The training curves illustrate the steep learning curve of TD3 and DR in the initial stages of learning, versus their robust counterparts. The M2TD3 agent ultimately achieves the highest performance at 5 million steps. Similarly, RARL exhibits a significant delay in learning, with stabilization occurring only toward the end of the training. Figures 4d and 4c show a significant variance in training across different random seeds. This emphasizes the difficulty of comparing different robust reinforcement learning methods along training.

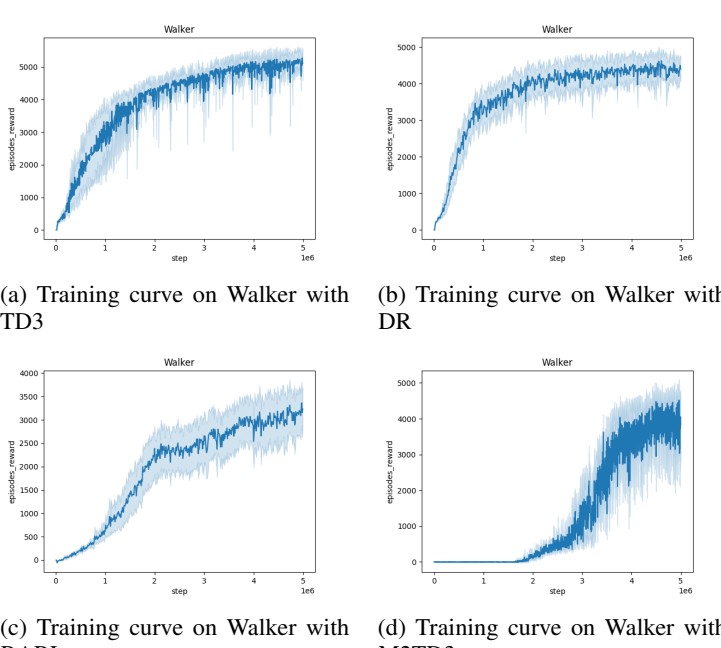

(a) Training curve on Walker with TD3

(b) Training curve on Walker with DR

(c) Training curve on Walker with RARL

(d) Training curve on Walker with M2TD3

Figure 4: Averaged training curves for Walker over 10 seeds

## 6 CONCLUSION

We introduced the Robust Reinforcement Learning Suite (RRLS), a Gymnasium-compatible benchmark for robust RL. RRLS standardizes evaluation across six MuJoCo tasks with predefined uncertainty sets and is simple to extend. We also provide four compatible baselines and static evaluations, enabling fair, reproducible comparisons. We hope RRLS serves as a dependable testbed that accelerates progress in robust reinforcement learning.

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

## A   APPENDIX

## A   MODIFIABLE PARAMETERS

The following tables list the parameters that can be modified in different MuJoCo environments used in the Robust Reinforcement Learning Suite. These parameters are accessed and modified through the `set_params` and `get_params` methods in the `ModifiedParamsEnv` interface.

| Parameter Name |
| --- |
| Torso Mass |
| Front Left Leg Mass |
| Front Left Leg Auxiliary Mass |
| Front Left Leg Ankle Mass |
| Front Right Leg Mass |
| Front Right Leg Auxiliary Mass |
| Front Right Leg Ankle Mass |
| Back Left Leg Mass |
| Back Left Leg Auxiliary Mass |
| Back Left Leg Ankle Mass |
| Back Right Leg Mass |
| Back Right Leg Auxiliary Mass |
| Back Right Leg Ankle Mass |

Table 6: Modifiable parameters from Ant environment

| Parameter Name |
| --- |
| World Friction |
| Torso Mass |
| Back Thigh Mass |
| Back Shin Mass |
| Back Foot Mass |
| Forward Thigh Mass |
| Forward Shin Mass |
| Forward Foot Mass |

Table 7: Modifiable parameters from Halfcheetah environment

| Parameter Name |
| --- |
| World Friction |
| Torso Mass |
| Thigh Mass |
| Leg Mass |
| Foot Mass |

Table 8: Modifiable parameters from Hopper environment

| Parameter Name |
|---|
| Torso Mass |
| Lower Waist Mass |
| Pelvis Mass |
| Right Thigh Mass |
| Right Shin Mass |
| Right Foot Mass |
| Left Thigh Mass |
| Left Shin Mass |
| Left Foot Mass |
| Right Upper Arm Mass |
| Right Lower Arm Mass |
| Left Upper Arm Mass |
| Left Lower Arm Mass |

Table 9: Modifiable parameters from Humanoid Stand Up environment

| Parameter Name |
|---|
| World Friction |
| Torso Mass |
| Thigh Mass |
| Leg Mass |
| Foot Mass |
| Left Thigh Mass |
| Left Leg Mass |
| Left Foot Mass |

Table 10: Modifiable parameters from Walker environment

| Parameter Name |
|---|
| Pole Mass |
| Cart Mass |

Table 11: Modifiable parameters from Inverted Pendulum environment

## B  TRAINING CURVES

We conducted training for each agent over a duration of 5 million steps, closely monitoring the cumulative rewards obtained over a trajectory spanning 1,000 steps. To enhance the reliability of our results, we averaged the performance curves across 10 different seeds.The graphs in Figures 5 to 8 illustrate how different training methods, including Domain Randomization, M2TD3, RARL, and TD3 impact agent performance across various environments.

## C  NON-NORMALIZED RESULTS

Table 12 reports the non-normalized worst case scores, averaged across 10 independent runs for each benchmark. Table 13 reports the average score obtained by each agent across a grid of environments, also averaged across 10 independent runs for each benchmark.

## D  IMPLEMENTATION DETAILS

### D.1  NEURAL NETWORK ARCHITECTURE

We employ the same neural network architecture for all baselines for the actor and the critic components. The architecture's design ensures uniformity and comparability across different models.

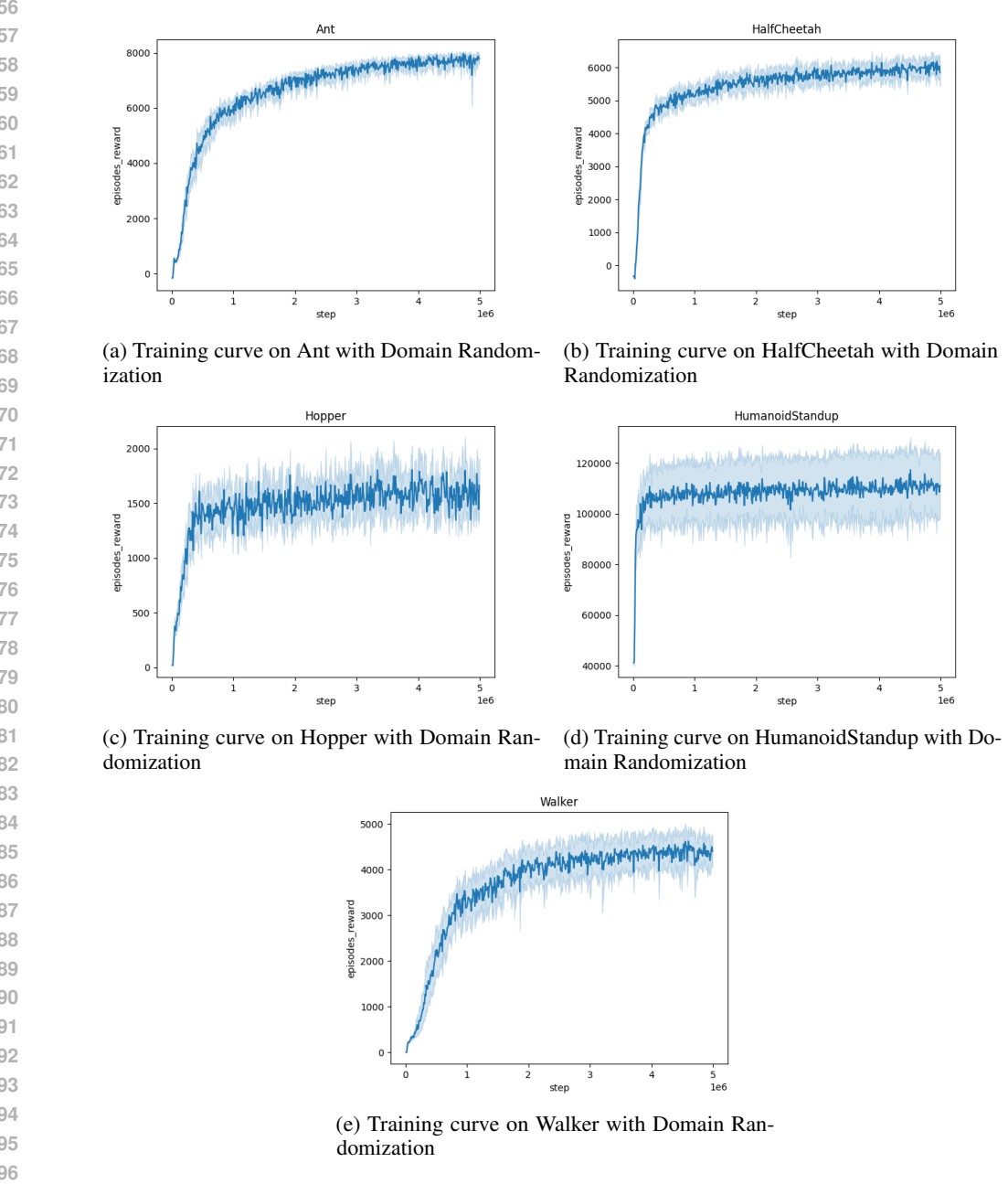

(a) Training curve on Ant with Domain Random-
ization

(b) Training curve on HalfCheetah with Domain
Randomization

(c) Training curve on Hopper with Domain Ran-
domization

(d) Training curve on HumanoidStandup with Do-
main Randomization

(e) Training curve on Walker with Domain Ran-
domization

Figure 5: Averaged training curves for the Domain Randomization method over 10 seeds

Table 12: Avg. of raw static worst-case performance over 10 seeds for each method

|        | Ant                  | HalfCheetah          | Hopper            | Humanoid StandUp          | Walker               |
|--------|----------------------|----------------------|-------------------|---------------------------|----------------------|
| DR     | $19.78 \pm 394.84$   | $2211.48 \pm 915.64$ | $245.01 \pm 167.21$ | $64886.87 \pm 30048.79$  | $1318.36 \pm 777.51$ |
| M2TD3  | $2322.73 \pm 649.3$  | $2031.9 \pm 409.7$   | $273.6 \pm 131.9$   | $71900.97 \pm 24317.35$  | $2214.16 \pm 1330.4$ |
| RARL   | $960.11 \pm 744.01$  | $-211.8 \pm 218.73$  | $170.46 \pm 45.73$  | $67821.86 \pm 21555.24$  | $360.31 \pm 186.06$  |
| NR-MDP | $-744.94 \pm 484.65$ | $-818.64 \pm 63.21$  | $5.73 \pm 8.87$     | $48318.45 \pm 11092.99$  | $16.42 \pm 3.5$      |
| TD3    | $-123.64 \pm 824.35$ | $-546.21 \pm 158.81$ | $69.3 \pm 42.77$    | $64577.24 \pm 16606.51$  | $114.41 \pm 211.05$  |

The critic network is structured with three layers, as depicted in Figure 9a, the critic begins with an
input layer that takes the state and action as inputs, then passes through two fully connected linear

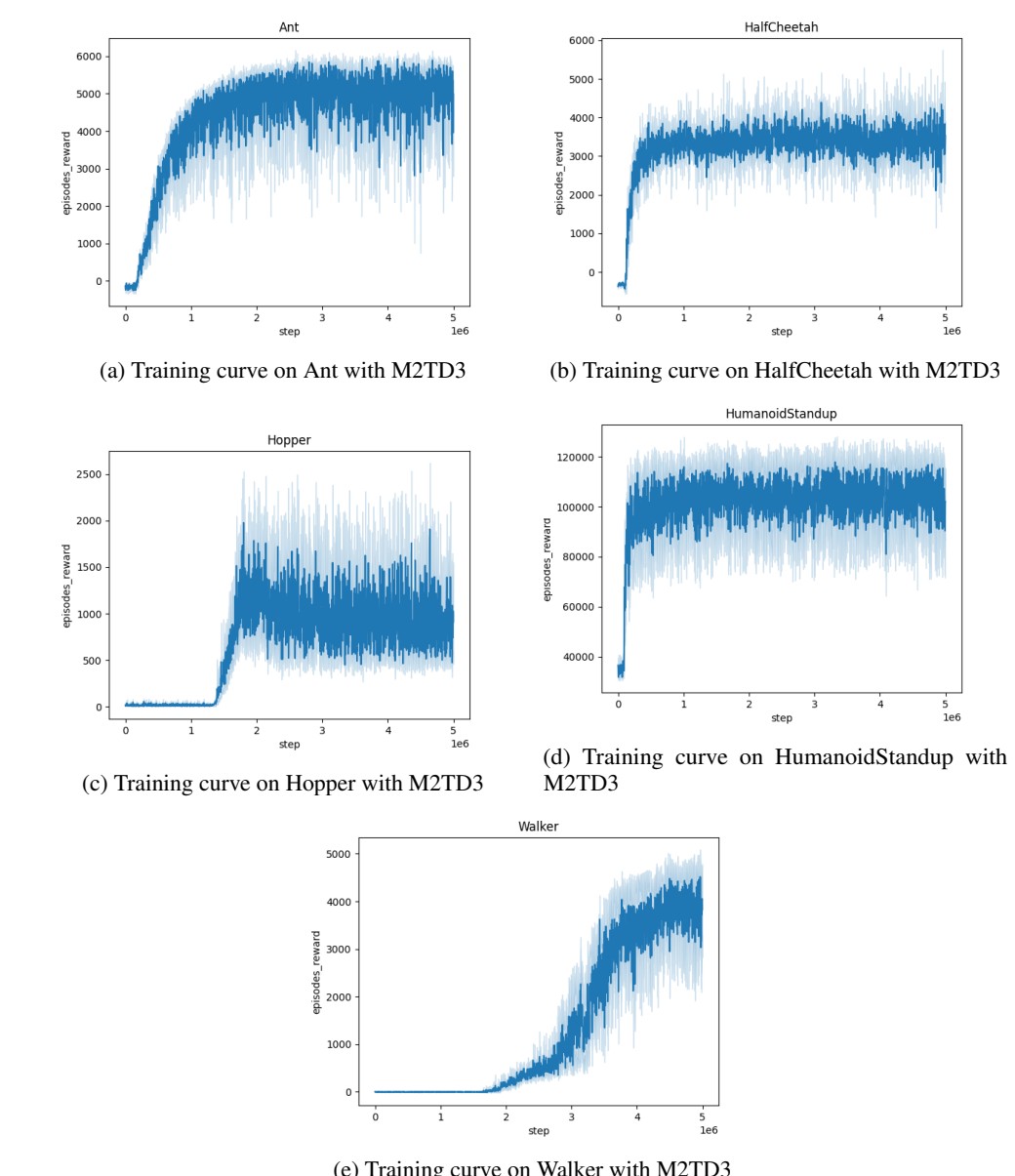

(a) Training curve on Ant with M2TD3

(b) Training curve on HalfCheetah with M2TD3

(c) Training curve on Hopper with M2TD3

(d) Training curve on HumanoidStandup with M2TD3

(e) Training curve on Walker with M2TD3

Figure 6: Averaged training curves for the M2TD3 method over 10 seeds

Table 13: Avg. of raw static average case performance over 10 seeds for each method

| env name algo-name | Ant | HalfCheetah | Hopper | Humanoid Standup | Walker |
|---|---|---|---|---|---|
| DR | $7500.88 \pm 143.38$ | $6170.33 \pm 442.57$ | $1688.36 \pm 225.59$ | $110939.89 \pm 22396.41$ | $4611.24 \pm 463.42$ |
| M2TD3 | $5577.41 \pm 316.95$ | $4000.98 \pm 314.76$ | $1193.32 \pm 254.9$ | $109598.43 \pm 12992.35$ | $4311.2 \pm 877.89$ |
| RARL | $4650.55 \pm 395.03$ | $206.71 \pm 887.25$ | $276.37 \pm 52.42$ | $104764.87 \pm 17400.85$ | $2493.26 \pm 1113.74$ |
| NR-MDP | $4197.80 \pm 90.66$ | $1388.90 \pm 283.25$ | $340.15 \pm 3.65$ | $92972.45 \pm 2251.18$ | $1501.05 \pm 453.96$ |
| TD3 | $2600.43 \pm 1468.87$ | $2350.58 \pm 357.12$ | $733.18 \pm 382.06$ | $100533.0 \pm 12298.37$ | $2965.47 \pm 685.39$ |

layers of 256 units each. The final layer is a single linear unit that outputs a real-valued function, representing the estimated value of the state-action pair.

The actor neural network, shown in Figure 9b, also utilizes a three-layer design. It begins with an input layer that accepts the state as input. This is followed by two linear layers, each consisting of

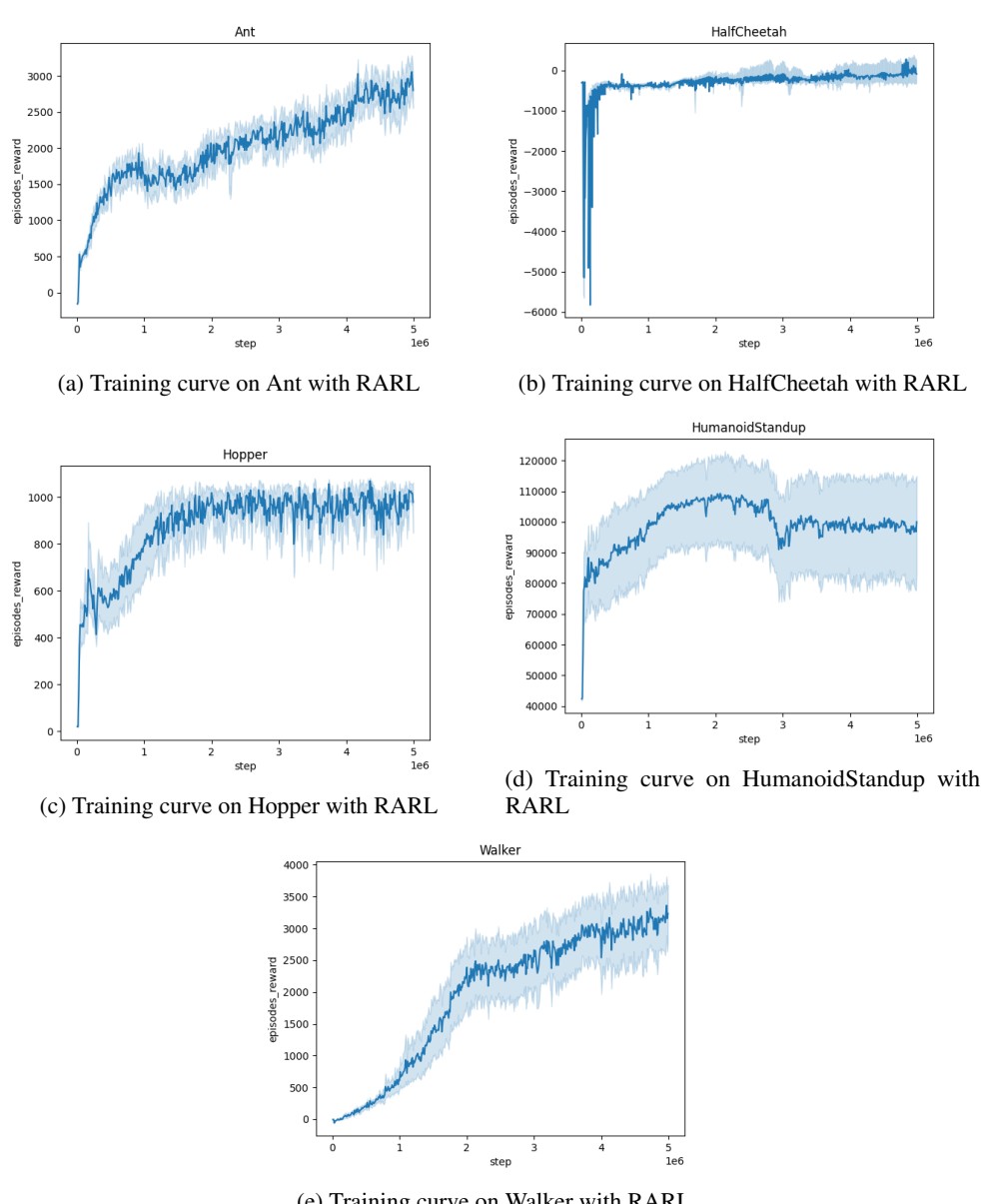

(a) Training curve on Ant with RARL

(b) Training curve on HalfCheetah with RARL

(c) Training curve on Hopper with RARL

(d) Training curve on HumanoidStandup with RARL

(e) Training curve on Walker with RARL

Figure 7: Averaged training curves for the RARL method over 10 seeds

256 units. The output layer of the actor neural network has a dimensionality equal to the number of dimensions of the action space.

## D.2 M2TD3

We use the official M2TD3 Tanabe et al. (2022) implementation provided by the original authors, accessible via the GitHub repository for M2TD3.

## D.3 TD3

We adopted the TD3 implementation from the CleanRL library, as detailed in Huang et al. (2022).

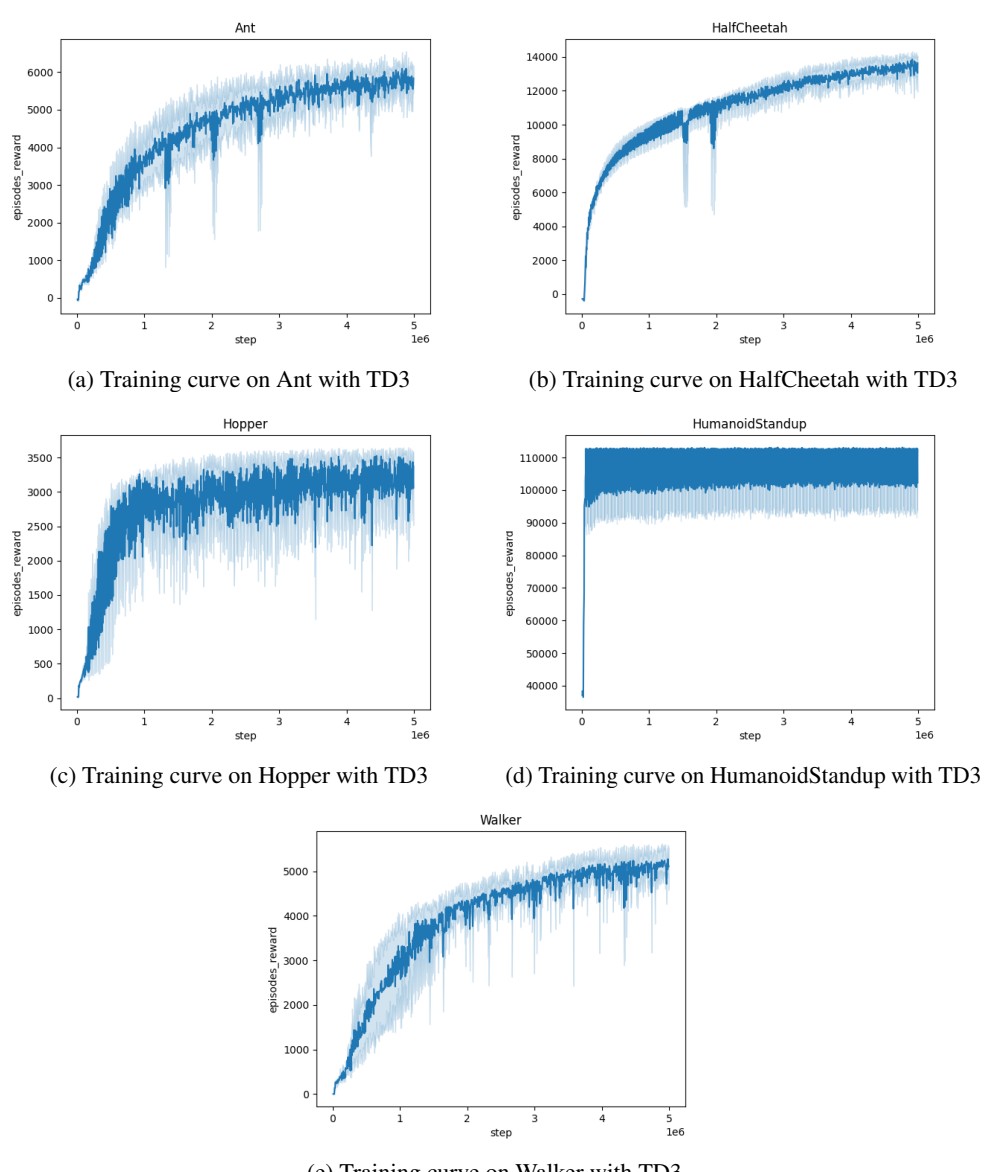

(a) Training curve on Ant with TD3        (b) Training curve on HalfCheetah with TD3

(c) Training curve on Hopper with TD3        (d) Training curve on HumanoidStandup with TD3

(e) Training curve on Walker with TD3

Figure 8: Averaged training curves for the TD3 method over 10 seeds

## E   Computer ressources

All experiments were run on a desktop machine (Intel i9, 10th generation processor, 64GB RAM) with a single NVIDIA RTX 4090 GPU. Averages and standard deviations were computed from 10 independent repetitions of each experiment.

## F   Broader impact

This paper proposes a benchmark for the robust reinforcement learning community. It addresses general computational challenges. These challenges may have societal and technological impacts, but we do not find it necessary to highlight them here.

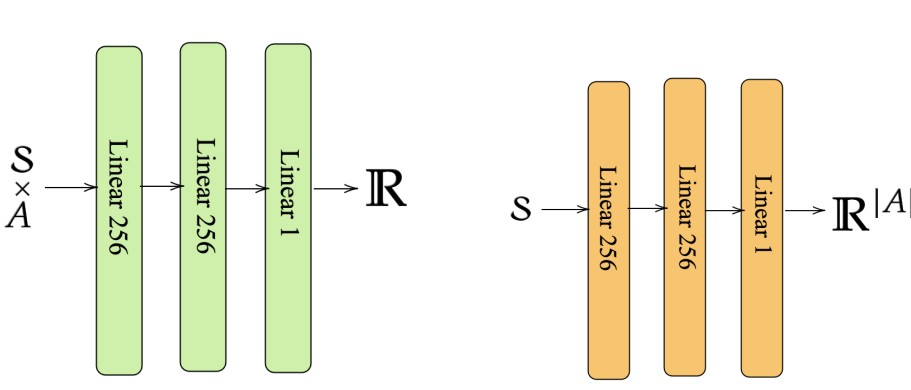

(a) Critic neural network architecture    (b) Actor neural network architecture

Figure 9: Actor critic neural network architecture

| Hyperparameter | Default Value |
|---|---|
| Policy Std Rate | 0.1 |
| Policy Noise Rate | 0.2 |
| Noise Clip Policy Rate | 0.5 |
| Noise Clip Omega Rate | 0.5 |
| Omega Std Rate | 1.0 |
| Min Omega Std Rate | 0.1 |
| Maximum Steps | 5e6 |
| Batch Size | 100 |
| Hatomega Number | 5 |
| Replay Size | 1e6 |
| Policy Hidden Size | 256 |
| Critic Hidden Size | 256 |
| Policy Learning Rate | 3e-4 |
| Critic Learning Rate | 3e-4 |
| Policy Frequency | 2 |
| Gamma | 0.99 |
| Polyak | 5e-3 |
| Hatomega Parameter Distance | 0.1 |
| Minimum Probability | 5e-2 |
| Hatomega Learning Rate (ho_lr) | 3e-4 |
| Optimizer | Adam |

Table 14: Hyperparameters for the M2TD3 Agent

| Hyperparameter | Default Value |
|---|---|
| Maximum Steps | 5e6 |
| Buffer Size | $1 \times 10^6$ |
| Learning Rate | $3 \times 10^{-4}$ |
| Gamma | 0.99 |
| Tau | 0.005 |
| Policy Noise | 0.2 |
| Exploration Noise | 0.1 |
| Learning Starts | $2.5 \times 10^4$ |
| Policy Frequency | 2 |
| Batch Size | 256 |
| Noise Clip | 0.5 |
| Action Min | -1 |
| Action Max | 1 |
| Optimizer | Adam |

Table 15: Hyperparameters for the TD3 Agent

