# OpenReview forum: "Rrls: Robust reinforcement learning suite"
_ICLR.cc/2026/Conference — Submitted to ICLR 2026_

### Official Review · Reviewer_MtfD · 2025-10-30

**Soundness:** 2
**Presentation:** 2
**Contribution:** 1
**Rating:** 2
**Confidence:** 3

**Summary:**

This paper introduces RRLS, a new benchmark suite for evaluating robust RL algorithms. RRLS provides six continuous-control tasks (Ant, HalfCheetah, Hopper, Walker, Humanoid Stand Up, and Inverted Pendulum) built on MuJoCo, each with two types of uncertainty sets (parameter perturbations and adversarial forces).

**Strengths:**

* The lack of standardized robust RL benchmarks is a well-known issue; this paper directly addresses it with a clear, well-scoped suite.

* The wrapper-based architecture makes RRLS easy to integrate with existing Gymnasium pipelines.

* The empirical section systematically compares multiple algorithms, highlighting trade-offs between robustness and average performance.

**Weaknesses:**

* There are some important related works providing robust RL benchmarks, that are missing in the related work and even the main motivations of the paper [1, 2].
* There exist other types of robustness perturbations, such as state, reward and action perturbations [3]. It would be helpful to claim the position of the paper compared with those works.

[1] Dulac-Arnold G, Levine N, Mankowitz D J, et al. An empirical investigation of the challenges of real-world reinforcement learning[J]. arXiv preprint arXiv:2003.11881, 2020.

[2] Gu S, Shi L, Wen M, et al. Robust gymnasium: A unified modular benchmark for robust reinforcement learning[J]. arXiv preprint arXiv:2502.19652, 2025.

[3] Moos J, Hansel K, Abdulsamad H, et al. Robust reinforcement learning: A review of foundations and recent advances[J]. Machine Learning and Knowledge Extraction, 2022, 4(1): 276-315.

**Questions:**

* Are training and testing environments generated by the same sampling distribution?
* Table 5: what is the meaning of the number after the robot name?

---

### Official Review · Reviewer_QAB7 · 2025-10-31

**Soundness:** 2
**Presentation:** 2
**Contribution:** 1
**Rating:** 2
**Confidence:** 4

**Summary:**

This paper provided additional functionalities mostly needed by robust RL works, i.e., changing parameters of the environmental dynamics such as friction, inertia, on top of the existing Gymnasium benchmark which was originally designed for more generic RL approaches.

**Strengths:**

* Paper is clearly written.
* A few baselines were included and benchmarked using the Rrls suite.

**Weaknesses:**

* The major concern from the reviewer is that there seems to already exist robust RL benchmarks, Robust Gymnasium (RG) [1], that could cover the setup considered by Rrls. Specifically, Rrls considered a fixed set of environmental disturbances (or the so-called uncertainty set in the paper), while RG considered a various types of disturbance.

* In general, Rrls maybe suited for a specific type of robust RL approaches, e.g., when the uncertainty set is fixed. However, there exist lines of robust RL research that are not specific to a pre-given set of uncertainties. For example, action perturbation [2, 3], risk-averse [4], or distribution robust [5].

* RG also seems to support more environments beyond the 6 supported by Rrls, and benchmarked against a wider suite of baselines.

[1] Gu, Shangding, et al. "Robust Gymnasium: A Unified Modular Benchmark for Robust Reinforcement Learning." The Thirteenth International Conference on Learning Representations.

[2] Stanton, Samuel, et al. "Robust reinforcement learning for shifting dynamics during deployment." (2021).

[3] Tessler, Chen, Yonathan Efroni, and Shie Mannor. "Action robust reinforcement learning and applications in continuous control." International Conference on Machine Learning. PMLR, 2019.

[4] Singh, Rahul, Qinsheng Zhang, and Yongxin Chen. "Improving robustness via risk averse distributional reinforcement learning." Learning for Dynamics and Control. PMLR, 2020.

[5] Liu, Zijian, et al. "Distributionally Robust $ Q $-Learning." International Conference on Machine Learning. PMLR, 2022.

**Questions:**

The reviewer is curious about the distinct advantages that RRLS offers compared to Robust-Gymnasium. Specifically, what unique benefits or experimental capabilities does RRLS provide that are not already covered by Robust-Gymnasium?

---

### Official Review · Reviewer_vEuz · 2025-10-31

**Soundness:** 2
**Presentation:** 3
**Contribution:** 2
**Rating:** 2
**Confidence:** 4

**Summary:**

The paper presents the Robust Reinforcement Learning Suite (RRLS), a Gymnasium-compatible benchmark designed to standardize robustness evaluation in continuous-control tasks. RRLS provides a clean API for parameter perturbations and defines non-rectangular uncertainty sets with multiple difficulty levels across six standard MuJoCo environments. The authors aim to establish a reproducible testbed for robust RL evaluation and demonstrate its use through experiments comparing representative algorithms.

**Strengths:**

RRLS offers a practical and well-structured interface for parameter perturbations within Gymnasium, consolidating previously scattered robustness protocols into a single reproducible pipeline. The paper documents the uncertainty sets with explicit parameter listings and ranges, enhancing comparability and experimental transparency. The paper includes baseline experiments on several standard algorithms under the proposed uncertainty sets, reporting training curves and seed-level performance variance as an empirical characterization of robustness behavior.

**Weaknesses:**

Relative to the current state of robust RL benchmarking, the contribution is narrow in scope. RRLS only covers six MuJoCo control tasks with environment-parameter uncertainty and does not address observation, action, or reward-level disruptions, nor broader domains such as safety, multi-agent, or vision-based RL. The benchmark lacks the breadth, modularity, and disruption coverage demonstrated in recent work like Robust-Gymnasium, which already defines a unified disrupted-MDP abstraction across multiple modalities.

The empirical evaluation is modest, reporting only a few standard baselines and offering limited analytical depth. No large-scale stress testing, cross-domain comparisons, or leaderboard-style benchmarking are included, making the study closer to a small-scale technical experiment than a community-standard benchmark.

Finally, the submission does not convincingly differentiate itself from existing platforms; its overlap with prior benchmarks undermines the novelty claim.

[1] Gu S, Shi L, Wen M, et al. Robust gymnasium: A unified modular benchmark for robust reinforcement learning[J]. arXiv preprint arXiv:2502.19652, 2025.
[2] Dulac-Arnold G, Levine N, Mankowitz D J, et al. An empirical investigation of the challenges of real-world reinforcement learning[J]. arXiv preprint arXiv:2003.11881, 2020.

**Questions:**

The paper does not clearly articulate its advantages over existing benchmark suites. What are the concrete improvements or distinctive features of RRLS, and is the code or framework publicly available?

---

### Official Review · Reviewer_ayZ2 · 2025-10-31

**Soundness:** 1
**Presentation:** 2
**Contribution:** 1
**Rating:** 2
**Confidence:** 5

**Summary:**

The paper proposes Robust Reinforcement Learning Suite (RRLS), a set of benchmark tasks to evaluate robustness of reinforcement learning (RL) algorithms. RRLS includes 6 continuous control MuJoCo tasks (Ant, HalfCheetah, Hopper, Humanoid Stand Up, Inverted Pendulum, Walker), with parametric uncertainty sets for robust evaluation on each task. The paper also provides experimental results for 5 algorithms (a non-robust RL algorithm, domain randomization, and 3 robust RL algorithms) across the 6 tasks for one of the proposed uncertainty sets.

**Strengths:**

**[S1]** Robust RL is an important area of research that could benefit from standardized benchmarks to evaluate existing algorithms and accelerate research advancements. The performance of robust RL algorithms often depends heavily on the choice of uncertainty set, and there is significant variation in the tasks / uncertainty sets used for evaluation across the field.

**Weaknesses:**

**[W1]** The authors claim there is a gap in benchmarks designed for robust RL, but they do not cite or discuss existing benchmarks that were designed to test robustness such as Robust Gymnasium [1] (ICLR 2025) and Real-World RL (RWRL) Suite [2]. Robust Gymnasium, in particular, provides a much more extensive set of tasks and disturbances for testing robustness in RL than the proposed RRLS, including the 6 tasks considered in RRLS. Given the existence of Robust Gymnasium, I do not believe this work provides a novel contribution to the field of robust RL.

**[W2]** The scope of the proposed benchmark set is very limited, containing only 6 basic continuous control MuJoCo tasks with parametric uncertainty sets for evaluation. It does not provide a wide range of tasks covering different domains, does not provide realistic robotics tasks such as the ones provided in Isaac Lab [3] and others, and does not provide a wide range of disturbance types for evaluation.

**[W3]** There is no analysis done to understand the difficulty levels of each environment contained in the proposed uncertainty sets. A standard RL algorithm should be trained on each specific evaluation environment to understand if meaningful performance can be achieved and quantify the trade-off required to achieve robustness. If an uncertainty set contains environments where meaningful performance is not possible, then it would not be a good choice for evaluating the performance of robust RL algorithms.

**[W4]** The paper only provides benchmark results for a single uncertainty set per task (the 3D uncertainty sets) under a single set of robustness hyperparameters for robust RL training. This provides minimal insight into the important performance vs. robustness tradeoff of robust RL algorithms under different training / evaluation setups. This limited experimental setup is insufficient for a paper focused on proposing a benchmark of tasks and uncertainty sets to advance robust RL research.

**References:**

[1] Gu et al., “Robust Gymnasium: A Unified Modular Benchmark for Robust Reinforcement Learning.” In ICLR 2025. https://github.com/SafeRL-Lab/Robust-Gymnasium

[2] Dulac-Arnold et al., “Challenges of real-world reinforcement learning: definitions, benchmarks and analysis.” Machine Learning, 2021. https://github.com/google-research/realworldrl_suite

[3] Mittal et al., “Orbit: A Unified Simulation Framework for Interactive Robot Learning Environments.” IEEE RA-L, 2023. https://github.com/isaac-sim/IsaacLab

**Questions:**

**[Q1]** Please discuss the contribution of this work compared to existing robust RL benchmark sets including Robust Gymnasium [1] and Real-World RL (RWRL) Suite [2]. Robust Gymnasium, in particular, already contains the 6 tasks proposed in this work as part of a comprehensive set of robust RL benchmark tasks. These related benchmark sets should also be cited and discussed in the paper.

**[Q2]** I recommend significantly expanding the experimental analysis to demonstrate why the provided choices of tasks and uncertainty sets represent a meaningful set of evaluation environments that will push forward the field of robust RL.

**[Q3]** How were the robustness hyperparameters of the robust RL methods selected / tuned? Please include these implementation details in the paper, as the performance of robust RL algorithms often depends heavily on the choice of these hyperparameters.

---

### Meta-Review · Area_Chair_ResL · 2026-01-07

**Summary:**

The submission introduces a benchmark based on Mujoco environments for robust reinforcement learning, named RRLS.

All reviewers share similar concerns:
1. Robust RL frameworks are not novel. There are already a few in the literature. The manuscript does not differentiate it from them.
2. The studied scope and scale are limited. The number of environments, the evaluated RL algorithms, and the considered robust types may be limited, as pointed out by different reviewers.
3. The analysis may not be in enough depth.

**Reviewer Concerns:**

Authors do not provide a rebuttal. Reviewers are unanimously negative. So I think the above concerns will persist till discussion eneds.

**Reviewer Scores:**

I think the reviewer scores are unlikely to change.

---

### Decision · Program_Chairs · 2026-01-26

Reject